# Experience of COVID-19 Vaccination among Primary Healthcare Workers in Hong Kong: A Qualitative Study

**DOI:** 10.3390/vaccines10091531

**Published:** 2022-09-15

**Authors:** Kai Man Ng, Tsun Kit Chu, Phyllis Lau

**Affiliations:** 1Department of Family Medicine & Primary Health Care, Tuen Mun Hospital, New Territories, Hong Kong, China; 2Department of General Practice, University of Melbourne, 780, Elizabeth Street, Melbourne, VIC 3010, Australia; 3School of Medicine, Western Sydney University, Locked Bag 1797, Penrith, NSW 2751, Australia

**Keywords:** COVID-19, vaccination, healthcare workers, primary care, experience, qualitative

## Abstract

Studies show that coronavirus disease 2019 (COVID-19) vaccine hesitancy exists among healthcare workers (HCWs). Past personal experiences of vaccination, such as the seasonal influenza vaccination, influence individuals’ intention to receive future vaccinations. This study aimed to explore the experience of COVID-19 vaccination among primary care HCWs in Hong Kong. A qualitative study using semi-structured interviews was conducted. Twenty-eight HCWs (ten doctors, ten nurses, and eight supporting staff) working in nine government-funded primary care clinics in Hong Kong who had completed at least one dose of COVID-19 vaccination were interviewed. Four themes were generated, namely, the cognitive and emotional battle of vaccine hesitancy, catalysts for vaccine acceptance, blasting vaccination myths, and being a positive influence. Providing timely, adequate, and transparent vaccine information and addressing the specific concerns of HCWs about the COVID-19 vaccine could enhance their vaccination uptake in future. Specific vaccine promotion strategies, such as the sharing of vaccination experiences targeted at different subgroups of HCWs, may improve vaccine acceptance through informational social influence.

## 1. Introduction

Coronavirus disease 2019 (COVID-19) vaccination is a key strategy to curb the pandemic. However, worldwide vaccine hesitancy during the roll out of various vaccines stunted uptake even among healthcare workers (HCWs). Vaccine hesitancy, thought to be vaccine specific, is defined as a delay in acceptance or refusal of vaccination despite an availability of vaccination services [1].

Vaccination of HCWs, who are at higher risk of acquiring infections in their routine clinical duties, is a well-tested infection-control measure in the prevention of nosocomial infections and a safeguard for exposed HCWs and patients, particularly patients who are immunocompromised [2,3]. A recently published wide-scope review of 26 papers that investigated the intentions and attitudes on COVID-19 vaccination of HCWs in 16 different countries across Africa, Asia, Europe (including Italy), and North America showed that COVID-19 vaccine hesitancy exists in HCWs, and acceptance of the COVID-19 vaccine among HCWs ranged from 27.7% to 92% in different populations [4]. In Hong Kong, an online survey performed in the early phase of the pandemic revealed that less than two-thirds of nurses intended to receive the COVID-19 vaccine [5]. However, with the implementation of various innovative strategies, more than 80% of HCWs working in the Hospital Authority (HA), the major provider of public healthcare services in Hong Kong, were eventually vaccinated by August 2021, around 6 months after the launch of the COVID-19 vaccination programme [6]. Other than mandatory vaccination, which had been shown to lack acceptance in an overseas study, the strategies initiated by the Hong Kong government and various organisations to promote COVID-19 vaccination included massive multi-media promotions, financial incentives such as lottery of private apartments, cars, gold, and gift vouchers, as well as disincentivizing measures such as imposing the requirement of regular SARS-CoV-2 antigen testing on unvaccinated individuals such as HCWs in the HA [7,8]. Understandably, many of these strategies were not evidence based, as they were driven by policy decisions made in response to the unique emergency and societal circumstances of the COVID-19 pandemic. This interesting phenomenon is not unique to Hong Kong; policy makers around the world reacted differently to the pandemic depending on their social and cultural contexts, COVID-19 vaccination coverage, and health promotion strategies used to promote vaccination. Rates of uptake differ markedly worldwide, regardless of vaccine availability.

A known determinant of vaccine hesitancy, in general, is past personal experience with vaccination [1]. A qualitative study conducted in Singapore revealed that previous positive experiences with influenza vaccination were one of the reasons why elderly people opted to get vaccinated again [9]. A survey across 56 countries on influenza vaccination similarly shows that personal vaccination experience significantly influenced HCWs’ decision for vaccination [10]. However, COVID-19 vaccine hesitancy is far more complex, and there is currently a lack of local research looking at the COVID-19 vaccination experience of HCWs or their attitudes towards the novel vaccine promotion strategies in Hong Kong.

This study aimed to explore the experiences of COVID-19 vaccination among primary care HCWs in Hong Kong to answer the primary research question, “What are primary HCWs’ experience with COVID-19 vaccination in Hong Kong?”, and sub-questions “What are their attitudes towards various vaccine promotion strategies?” and “What are their perceived facilitators and barriers towards COVID-19 vaccination?”.

## 2. Methods

### 2.1. Design

This was a qualitative study using a phenomenological approach and semi-structured interviews.

### 2.2. Research Team

The research team consisted of KN, TC, and PL. KN and TC are both primary care doctors working in the New Territories West region in Hong Kong. PL is an experienced primary care qualitative researcher. The team reflected on and reconciled any potential professional biases regularly when designing the interview questions, recruiting participants, analysing the data, and writing the report and manuscript.

### 2.3. Research Setting

The research was conducted across nine public primary care clinics in the New Territories West region in Hong Kong. This region has a population of around 1.1 million, one-seventh of the whole Hong Kong population [11]. Patients in this region generally have a lower socio-economic class. The nine clinics provided 872,000 consultations in 2016 [11].

### 2.4. Participant Recruitment

HCWs working in nine public primary care clinics in the New Territories West region of Hong Kong who had received at least one dose of COVID-19 vaccination were purposively recruited by KN to the study through email and invitation letters from 1 December 2021 to 31 March 2022. Participants were selected using a matrix of demographic characteristics such as age, gender, profession, training, and working experience in order to achieve maximum variation. Written informed consent was obtained from all participants.

### 2.5. Data Collection

An interview guide covering the factors affecting vaccination, vaccination experience, effects of vaccination, and attitudes towards vaccination promotion strategies was developed based on the 5C psychological antecedents of vaccination (confidence, complacency, constraints, calculation, and collective responsibility), and was listed in Table 1 [12]. Confidence is defined as trust in vaccines, the system that delivers them, and the motivations of policy-makers who decide on the need for vaccination [1]. Complacency “exists where perceived risks of vaccine-preventable diseases are low and vaccination is not deemed a necessary preventive action” [1]. Constraints refer to “physical availability, affordability and willingness-to-pay, geographical accessibility, ability to understand and appeal of immunization service affecting uptake” [1]. Calculation refers to individuals’ engagement in extensive information searching [12]. Collective responsibility is defined as the “willingness to protect others by one’s own vaccination by means of herd immunity” [12].

Researcher KN conducted all in-depth, semi-structured, one-to-one, and face-to-face interviews with participants using the interview guide. Interviews were conducted in the staff rooms of clinics with adequate privacy (only the interviewer and interviewee were present) outside of the service hours of the clinics.

All interviews were conducted in KN and the participants’ mother tongue (Cantonese) and audio-recorded with prior consent from participants. Field notes were taken during the interviews. Interviews were transcribed verbatim and identifiers such as names were removed. Member checking was performed, i.e., transcripts were provided to participants for verification to enhance the integrity of the data collection process. Recruitment and interviews continued until data saturation was determined to be reached when no new information emerged from the interviews.

### 2.6. Data Analysis

Transcripts were imported into NVivo 12 and analysed thematically using a mixed inductive and deductive approach based on the framework proposed by Braun and Clarke [13,14]. Transcripts were analysed in Cantonese. They were first coded inductively by researcher KN. Cross-coding was performed independently by researcher TC. Discrepancies in coding were identified and discussed between KN and TC until a consensus was reached. This process also included subgroup analysis based on the participants’ professions, as well as analysis of merged data. Themes were then deductively elicited using the 5C psychological antecedents of vaccination as a framework. Salient quotes to substantiate the themes were then translated into English by KN and back-translation was performed by TC to ensure integrity of translation.

### 2.7. Ethical Consideration

This study was approved by the New Territories West Cluster Research Ethics Committee (reference number NTWC/REC/21075) on 1 November 2021.

To minimise the potential influence of power difference or superior–subordinate relationship between KN (who is a doctor) and participants, KN clearly stated his neutral stance (i.e., not representing any administrative or managerial staff), assured strict confidentiality, and reiterated that the data collected were solely for research purposes and would not influence their relationship with researchers, their career, and advancement.

## 3. Results

A total of 28 participants were recruited. Ten doctors (one consultant, three associate consultants, one resident specialist, three resident trainees, and two service residents), ten nurses (one advanced practice nurse and nine registered nurses), and eight supporting staff (two clinic clerks, five patient care assistants, and one phlebotomist) were interviewed. The demographics of the individual participants are listed in Table 2. The predominantly female nurses and supporting staff participants were consistent with the demographic makeup of these two groups in Hong Kong [5]. Nurse participants generally had a longer work experience, and doctor participants generally received their COVID-19 vaccinations earlier.

Four main themes, namely the “cognitive and emotional battle of vaccine hesitancy”, “catalysts for vaccine acceptance”, “blasting myths of vaccination” and “being a positive influence” were elicited from the data and categorised into two categories: “before vaccination” and “after vaccination” [Figure 1].

### 3.1. Theme 1: Cognitive and Emotional Battle of Vaccine Hesitancy

#### 3.1.1. Perceived COVID-19 Susceptibility

Most participants perceived themselves to be prone to infections because of frequent patient interactions in primary care. Generally, they agreed that they needed vaccination to prevent and reduce the severity or complications of infections. Moreover, they believed vaccination could protect people they care about, such as family, patients, colleagues, the health care system, and even the city of Hong Kong.

“*I’m very concerned (of infection) at the moment, because of the Omicron variant and its severe outbreak in other countries. It seems we could not escape from it. …absolutely I need to vaccinate to prevent infection.*”(Participant N8, 48-year-old, female, advanced practice nurse, 24 years of working experience)

“…*healthcare workers should protect our patients, friends, and family members we encounter in daily life, and even more, if all of us are vaccinated, then we could protect our community, and the pandemic would end sooner*.”(Participant D8, 30-year-old, female, GP resident trainee, 5 years of working experience)

#### 3.1.2. Scepticism towards Vaccine

Although participants thought they should vaccinate, many concerns caused hesitation and stopped them from doing so. In particular, junior HCWs such as supporting staff and some nurses seemed more likely to have such scepticisms. Concerns were expressed about the number, scale, and standards of the clinical trials conducted, and therefore, the trustworthiness of the COVID-19 vaccine research and evidence. Expedited production and approval for emergency use of the COVID-19 vaccines discounted participants’ confidence. Vaccines produced by branded pharmaceutical companies seemed to be more trusted. Innovative vaccine platforms such as the mRNA vaccine were new to people in Hong Kong and some participants questioned their use. Occurrence of infection, even after three doses of the vaccine, reinforced participants’ doubts about the vaccines’ efficacy. Frequent reporting of post-vaccination adverse events in the news exacerbated fear and anxiety about safety.

“*I’m afraid, fear the side effects, because it did not follow conventional procedures and put into market.*”(Participant S2, 35-year-old, male, clerical staff, 10 years of working experience)

“*I took [deidentified] (an inactivated COVID-19 vaccine), which uses an established technology, … unlike [deidentified] (an mRNA COVID-19 vaccine) which is very new and there are lots of uncertainties*.”(Participant N3, 45-year-old, female, registered nurse, 21 years of working experience)

“*I think the vaccine is useless, as you can see people were still infected after receiving three doses of vaccine*.”(Participant N6, 51-year-old, female, registered nurse, 26 years of working experience)

“…*when the news reported that there were a lot of people got severe side effects, it stopped me from vaccination. …those sudden death, myocarditis during exercises, and stroke... It was in the news everyday*.”(Participant N4, 32-year-old, female, registered nurse, 9 years of working experience)

#### 3.1.3. Autonomy

Participants valued the autonomy that they currently have and believed their decisions about whether to vaccinate should be respected.

“*We have autonomy, especially healthcare workers. Each of us has our judgement on vaccination, to balance risks and benefits, and it should not be mandatory*.”(Participant D2, 30-year-old, female, GP resident trainee, 7 years of working experience)

#### 3.1.4. Source of Information

Information about the vaccines was inadequate, especially at the start of the vaccination programme. While doctors mostly obtained their information about the COVID-19 vaccines from primary sources such as peer-reviewed journals and manufacturers’ product information, most nurses and supporting staff obtained their information from secondary sources such as the media, internet, friends, and relatives. Some doctors noted the phenomenon and suggested that bespoke information for nurses and supporting staff should be designed. Doctors were perceived by nurses and supporting staff as reliable sources of information.

“*In March 2021, there were many scientific papers published which were large-scale studies, especially the [deidentified international] study. I obtained information from those studies.*”(Participant D10, 48-year-old, male, GP consultant, 23 years of working experience)

“*I really had to find (information about the vaccines) by myself, as we didn’t have enough information*.”(Participant S1, 55-year-old, female, patient care assistant, 10 years of working experience)

“*I had (enough information about the vaccines), from news and my friend.*”(Participant S8, patient care assistant, 59-year-old, female, 10 years of working experience)

“*Frontline staff such as nurses and supporting staff may not have update from the latest study. Perhaps one could publish some pamphlets and newsletter to enhance their understanding on the disease and vaccines, so that they can make an appropriate choice*.”(Participant D6, 42-year-old, male, associate GP consultant, 19 years of working experience)

### 3.2. Theme 2: Catalysts for Vaccination

#### 3.2.1. Informational Social Influence

Participants who were hesitant about vaccination adopted a watchful waiting approach. The experience of other vaccine recipients was the most common factor reported by participants, particularly nurses and supporting staff, to influence their decision to vaccinate or not. Stories of negligible side effects from the COVID-19 vaccines helped relieve some participants’ anxieties, motivating them to receive the vaccine. Some HCWs were inspired by well-known celebrities and government officials in their decision to receive the vaccine.

“*I observed other people for 7 months before I had the confidence to vaccinate*.”(Participant N2, 46-year-old, female, registered nurse, 19 years of working experience)

“*I planned to vaccinate once available, but then I changed my mind and observed first. When I noticed half of colleagues in my clinic got vaccinated, and they did not have many side effects, then I decided to vaccinate*.”(Participant D8, 30-year-old, female, GP resident trainee, 5 years of working experience)

“*Government officials and celebrities vaccinated first, to be a role model…is effective*.”(Participant S7, 56-year-old, female, phlebotomist, 15 years of working experience)

#### 3.2.2. Ease of Access

Reducing practical barriers to vaccination motivated participants to vaccinate, such as establishing vaccination venues in proximity to work locations, providing transportation to vaccination venues, and providing vaccination to staff at work.

“*It is convenient for me to vaccinate at [deidentified] Hospital (close to his workplace). If I vaccinated earlier, I would have to travel to community vaccination centre which was troublesome*.”(Participant D2, 30-year-old, female, GP resident trainee, 7 years of working experience)

#### 3.2.3. Social Responsibility

Doctor participants generally received vaccines soon after the launch of the vaccination programme. A belief in social responsibility was common among the doctor participants.

“*Social responsibility (is a factor affecting vaccination), because I am a medical professional. I have to do it myself first in order to be role model for others especially the public. We are an important symbolic figure, and we should get the shot first in order to motivate them.*”(Participant D10, 48-year-old, male, GP consultant, 23 years of working experience)

#### 3.2.4. Incentives

In order to boost HCWs’ vaccination rates, several incentives were offered by the Hospital Authority, including provision of authorised vaccination leave (AVL) and financial rewards such as lottery of smartphones, smartwatches, tablets, shopping coupons, and amusement park tickets. Generally, HCWs welcomed these measures, although it seemed to motivate nurses and supporting staff to vaccinate more than doctors. A number of participants appreciated AVL because it allowed them to rest after vaccination. However, they thought that the duration and flexibility of the AVL could be improved. Financial rewards were inferior to AVL in terms of attractiveness. Some participants explained that financial reward was not the most important factor affecting their decision, and some even expressed concern that these strategies would distort the fundamental principle of vaccination.

“*I vaccinated (in June 2021) because there was AVL.*”(Participant S3, 23-year-old, female, clinic clerk, 2 years of working experience)

“*The most important thing we concern about (the vaccines) is the efficacy in protection. Financial reward is just an additional benefit*.”(Participant D7, 39-year-old, female, GP resident, 16 years of working experience)

“*I don’t suggest using lucky draw to promote vaccination. It seems like you vaccinate because you want to get the prize, but not protection*.”(Participant N3, 45-year-old, female, registered nurse, 21 years of working experience)

#### 3.2.5. Perverse Incentives

It became the policy of local public healthcare sector in the later phase of the vaccination programme that unvaccinated HCWs were required to undergo regular COVID-19 polymerase chain reaction (PCR) tests at their own expense. Several participants admitted that it became the impetus to vaccinate. Some HCWs experienced stress when their supervisors arranged individual meetings with them and asked for their reasons for not vaccinating. Yet, this turned out to be a motivator for them to vaccinate in order to avoid further counselling by their superiors. Social restrictions imposed on unvaccinated individuals by the Hong Kong Government, such as prohibition from access to certain places, and limitations on the number of people dining together, though not targeted at HCWs, also drove our participants to get vaccinated.

“…*and if I did not vaccinate by 1 September, we had to pay for the test. That’s a big concern and I decided to vaccinate*.”(Participant N4, 32-year-old, female, registered nurse, 9 years of working experience)

“*Yes, I did not vaccinate by June and therefore my supervisor asked me, when I would vaccinate, why I haven’t vaccinated, and I felt stressed. However, since I got the jab, there was no more stress*.”(Participant N10, 29-year-old, female, registered nurse, 7 years of working experience)

“*I vaccinated because of social convenience. If you don’t vaccinate, you can’t do a lot of things, such as entering some public venues*.”(Participant N2, 46-year-old, female, registered nurse, 19 years of working experience)

#### 3.2.6. Divided Views on Mandatory Vaccination

Although mandatory vaccination was not implemented in Hong Kong, participants were asked about their views on the controversial policy. Participants had divided opinions. Perceived severity of the pandemic influenced their views. Participants of managerial level seemed more likely to support mandatory vaccination and reasons included suboptimal vaccination rate despite government and hospital incentives and HCWs’ obligation to protect patients and the healthcare system and be role models to the public. On the other hand, some participants believed that the autonomy of HCWs should be respected and that vaccination may not be appropriate for everyone.

“*The aim (of mandatory vaccination) is to fight the pandemic together. We are in the medical field, and if we don’t take the first step, then how can we lead our citizens? Therefore, I think it is acceptable*.”(Participant N8, 48-year-old, female, advanced practice nurse, 24 years of working experience)

“*Some colleagues are physically unfit for vaccination, and therefore you cannot make it a mandate. Do you count it as injury on-duty if adverse effects happen after vaccination?*”(Participant N3, 45-year-old, female, registered nurse, 21 years of working experience)

### 3.3. Theme 3: Blasting Myths of Vaccination

Vaccinations were thought by many HCWs to be an imposition, but most participants had positive vaccination experiences that were different to their assumptions, e.g., the booking process was simpler and quicker than they imagined. Most received their vaccinations at government-run community vaccination centres and some at public hospitals. They were satisfied with appointment booking, simple registration process, comfortable environment, clear instructions, and the smooth and fast vaccination process.

Participants also reported that the side effects after vaccination were far milder than what they were made to believe. Common side effects reported were injection site pain, chills, fever, headache, malaise, and myalgia, which were all clearly detailed in manufacturers’ product information. The experienced side effects were short-lived and had minimal impact on their daily routines. Some participants commented on the misinformation about the side effects of COVID-19 vaccine amongst the general public.

“*The process was smooth. Basically, the waiting time was short, and I did not have much paperwork to do, and therefore, after the staff scanned my QR code and checked my name, I was ready to vaccinate*.”(Participant D5, 42-year-old, male, associate GP consultant, 19 years of working experience)

“*I had some malaise and some injection site pain, and there was no fever. I worked and exercised as usual. I didn’t know there could be myocarditis at that moment, and I just kept running and hiking without any problem*.”(Participant D10, 48-year-old, male, GP consultant, 23 years of working experience)

“*The side effects weren’t as severe as circulated in public.*”(Participant N2, 46-year-old, female, registered nurse, 19 years of working experience)

### 3.4. Theme 4: Being a Positive Influence

Participants believed that vaccination did not only not have long-term or severe sequalae to their health, but it even positively impacted their mental and social wellbeing. They felt more protected and safer at work and their social lives became less restricted by social distancing measures. Many participants actively shared their positive experiences with colleagues and patients, and recommended vaccination to the unvaccinated. Positive experiences with the first two doses of vaccination also motivated participants to receive the subsequent booster dose without hesitation.

“*I felt less feared (after vaccination). Previously I was concerned about infection and complications (of COVID-19), and now it was prevented*.”(Participant D2, 30-year-old, female, GP resident trainee, 7 years of working experience)

“…*when I know my colleagues and friends aren’t vaccinated, I’ll remind them about the urgency of vaccination, as the most important reason for vaccination to protect themselves*.”(Participant S5, 41-year-old, female, patient care assistant, 6 years of working experience)

“*(After vaccination myself,) I am more confident in encouraging patients in getting jabs.*”(Participant S1, 55-year-old, female, patient care assistant, 10 years of working experience)

“*I took the booster dose much earlier because the last 2 doses were unremarkable.*”(Participant N7, 51-year-old, female, registered nurse, 25 years of working experience)

## 4. Discussion

To the best of our knowledge, this is the first qualitative study conducted in Hong Kong exploring the experience of COVID-19 vaccination among primary HCWs and their attitudes towards the novel vaccine promotion strategies in Hong Kong. Our study revealed that HCWs generally perceived themselves to be susceptible to COVID-19 infection and therefore requiring vaccination. However, they are hindered by various concerns.

Informational social influence is an important trigger identified in this study for the decision to vaccinate. According to American Psychological Association, informational social influence refers to the interpersonal processes that challenge the correctness of an individual’s belief or the appropriateness of his or her behaviour, thereby promoting changes [15]. After observing the people around them receiving the vaccination, participants gradually changed their mind and followed their example. It is of interest that the phenomenon exists among HCWs who are supposedly better informed compared to the general public. This is consistent with findings in a multi-national online cross-sectional survey in 2021. In that study, action cues such as seeing others in the community receiving the vaccination and doctors recommending the vaccination were positively associated with COVID-19 vaccination intention [16]. Another study in the US showed that certain action cues such as recommendation from personal providers and beliefs of friends, family, colleagues, and supervisors that one should vaccinate motivated HCWs to receive COVID-19 vaccines [17]. Therefore, vaccination promotion in HCWs could leverage this psychological phenomenon and target individuals with strong social influence to vaccinate early and exert positive social pressure on others.

Concerns about the COVID-19 vaccines expressed by the HCWs in our study, especially by the supporting staff, resonate with findings from other literature. A survey focused on HCWs choosing not to vaccinate found that 69.6% of respondents expressed concerns about vaccine side effects [18]. A qualitative study in the UK found that junior HCWs were more likely to have concerns about COVID-19 vaccine safety [19]. Many of these concerns were caused by inadequate information, misinformation, and mistrust in the source of information. High-quality, accurate, and easy-to-understand information was not readily available, even to HCWs. Inadequate information could cause an individual to adopt watchful waiting approach, as found in a US qualitative study [20]. HCWs’ concerns about side effects should be addressed directly and promptly by timely, concise, and digestible vaccine information before they evolve into scepticism and further encroaches confidence towards vaccine and aggravate hesitancy.

In our study, a number of participants’ decisions to vaccinate were heavily influenced by government policies or organisational regulatory measures put in place in Hong Kong to increase COVID-19 vaccination. This is comparable to existing research findings. In a systematic review of the effectiveness of different campaign strategies for influenza vaccination, strategies were classified into education and promotion, incentives (such as prizes and free-of-charge vaccination), and organisational strategies and policies (such as vaccinate-or-wear-a-mask and mandatory vaccination) [21]. The review showed that regulatory measures were the most effective to effect an increase in vaccination coverage (VC) percentage, while mandatory vaccination achieved the highest VC. Incentives only provided a modest increase in VC when used alone. Vaccination promotion is a multi-dimensional process and should start with public health education and readily available accurate information. Incentives could be added as motivation to combat vaccine hesitancy. Regulatory measures should be reserved as a last resort when VC is suboptimal in spite of existing initiatives. There are currently no mandatory measures enforced in Hong Kong, and HCWs and the general public have a right to decide whether to vaccinate. Mandatory vaccination should not be implemented unless there is consensus among stakeholders after thorough consideration of risks, benefits and ethical aspects.

Positive vaccination experience significantly altered participants’ perception and subsequent behaviour. Most participants recommended vaccines to others after they had received one. This is consistent with the findings of other studies. A mixed-method study conducted in California in 2021 found that vaccinated HCWs believed it was their responsibility to be role model for their patients and shared their personal vaccination experience [22]. Furthermore, positive vaccination experience seems to also have a positive impact on our participants’ intention and willingness to receive booster doses. It is reasonable to anticipate that further booster doses may be required in the future and HCWs’ previous positive vaccination experience could facilitate faster uptake.

Although this study focused on understanding the COVID-19 vaccination experience of HCWs, our findings suggest several potential strategies to enhance uptake of vaccinations in general among HCWs in Hong Kong. Vaccination promotional campaigns could start with HCWs such as doctors who are regarded as more medically knowledgeable. They could, in turn, be role models or ambassadors and actively share their experience, as they are regarded as credible sources of information and would enhance the confidence of other HCWs [16]. Adequate, transparent, and balanced information should be provided to HCWs to gain trust [21,23]. Regulatory measures, though not preferred, could be enforced at later stages when other initiatives failed to improve uptake.

There are limitations to our study. Recall bias was unavoidable as this study required participants to retrospectively recount their experience. The rapidly-changing nature of the COVID-19 pandemic in Hong Kong had meant that vaccination promotion was extremely dynamic, and different strategies were enforced at different timepoints during the study. Not all participants experienced the same strategies and earlier interviews did not cover views on strategies that were implemented later. The study recruited HCWs from the public primary health care sector in Hong Kong and only those who were vaccinated. This may limit generalisability. Future research in the private sector would be necessary. To date, very few HCWs in Hong Kong are unvaccinated, but investigating their attitude and perceptions would also be important to shed light on COVID-19 vaccine hesitancy.

## 5. Conclusions

Vaccination is a critical measure to end the COVID-19 pandemic. It is important to achieve a high vaccination rate in HCWs, not only because they have a high exposure risk, but because they also have an important and unique role in building the confidence of the general public towards COVID-19 vaccines. Our study revealed informational social influence is an important trigger for the decision to vaccinate among HCWs, Moreover, vaccination promotion requires timely, adequate, and transparent information, and addressing the specific concerns of HCWs about the COVID-19 vaccines would enhance their vaccination uptake. HCWs’ concerns about the side effects may evolve into scepticism and encroach confidence towards vaccination if no tailor-made information is provided. In spite of the fact that COVID-19 vaccination uptake among HCWs at present is high in many countries, our study findings provide evidence to guide implementation of policies for COVID-19 vaccination programmes. Specific strategies leveraging on informational social influences by targeting individuals or HCWs with strong social influence to vaccinate early may improve vaccine acceptance in some subgroups of HCWs. At the same time, efforts should be made to combat the misinformation and mistrust in the sources of information in the healthcare setting, as well as in the community. Future research is needed to determine the effectiveness of vaccination information delivery strategies to HCWs and how they could be enhanced.

## Figures and Tables

**Figure 1 vaccines-10-01531-f001:**
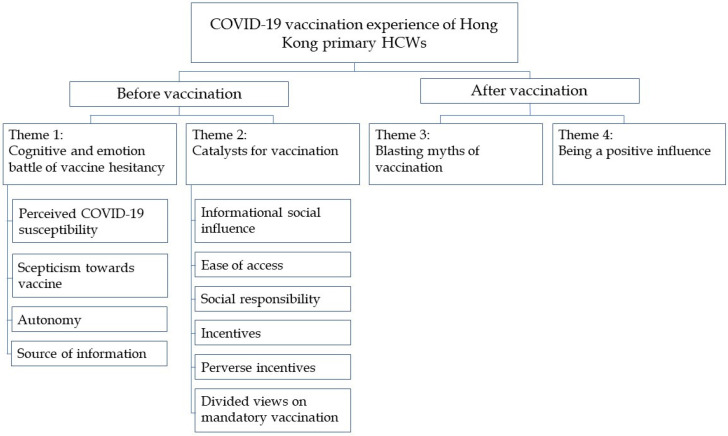
Themes and subthemes of the study.

**Table 1 vaccines-10-01531-t001:** Interview guide.

Topic	Question(s)	Probes/Prompts
Introduction	What comes to mind when you first hear the term “COVID-19 vaccines”?	
Factors affecting vaccination	Please tell me the factors affecting your intention to vaccinate.	How at risk did you think you were of contracting COVID-19? Considering your risk, did you think you were indicated for COVID-19 vaccination? Please elaborate.
		Did your confidence in COVID-19 vaccine affect your decision to vaccinate? Please elaborate.
		Were there any barriers for you to receive a COVID-19 vaccines? Please elaborate.
		Did you feel you have adequate information to decide whether to have the COVID-19 vaccine? If not, what sort of information did you lack? Please elaborate.
		Some people vaccinate because they want to protect others. What is your view?
		Are they any other reasons that affect
	When did you decide to receive the COVID-19 vaccine? How did you make up your mind at that moment?	
Vaccination experience	Please tell me your experience of receiving the vaccination.	Where were you vaccinated?
		How did you make the appointment?
		Did you vaccinate during or outside work hours?
		How did you feel about the arrangement at your vaccination venue?
		What suggestions do you have for improving the vaccination experience?
Effects of vaccination	Please tell me your experience after vaccination.	Did you experience any side effects? If yes, what are they and how long did they last?
	Did vaccination change your daily life? If yes, how	Did vaccination have impact on your work?
		Did vaccination have impact on your social life?
		Did vaccination have impact on your health?
	Considering your experience, would you recommend COVID-19 vaccination to others?	
Attitudes on vaccination promotion strategies	What vaccination promotion strategies do you know that targeted healthcare workers in Hong Kong? What are your views on these strategies?	What are your views on promotion from social media/authorised vaccination leave/lucky draw for vaccinated people?
	Some countries employed mandatory vaccination policies for healthcare workers. What is your opinion on such strategy?	
	Do you have any suggestion for vaccination promotion strategy targeting healthcare workers?	

**Table 2 vaccines-10-01531-t002:** Demographics of individual participants.

Participant ID	Ranking	Gender	Age	Working Experience (Years)	Time between Vaccination and Launch of Vaccination Programme (Months)
D1	GP resident specialist	Male	39	14	0
D2	GP resident trainee	Female	30	7	3
D3	GP associate consultant	Male	44	19	0
D4	GP resident	Male	52	29	1
D5	GP associate consultant	Male	42	19	0
D6	GP associate consultant	Male	42	19	0
D7	GP resident	Female	39	16	2
D8	GP resident trainee	Female	30	5	1
D9	GP resident trainee	Female	28	5	1
D10	GP consultant	Male	48	23	0
N1	Registered nurse	Female	44	19	4
N2	Registered nurse	Female	46	19	7
N3	Registered nurse	Female	45	21	2
N4	Registered nurse	Female	32	9	4
N5	Registered nurse	Female	64	42	4
N6	Registered nurse	Female	51	26	4
N7	Registered nurse	Female	51	25	0
N8	Advanced practice nurse	Female	48	24	1
N9	Registered nurse	Female	32	9	1
N10	Registered nurse	Female	29	7	4
S1	Patient care assistant	Female	55	10	3
S2	Clinic clerk	Male	35	10	3
S3	Clinic clerk	Female	23	2	3
S4	Patient care assistant	Female	46	10	3
S5	Patient care assistant	Female	41	6	3
S6	Patient care assistant	Female	59	19	3
S7	Phlebotomist	Female	56	15	0
S8	Patient care assistant	Female	59	10	3

## Data Availability

The data presented in this study are available upon reasonable request to the corresponding author.

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
