# Peer review of "Experience of COVID-19 Vaccination among Primary Healthcare Workers in Hong Kong: A Qualitative Study"

_vaccines, 2022, doi:10.3390/vaccines10091531_

Round 1

Reviewer 1 Report

Thank you for the invitation. This is an interesting study. Other quantitative studies could not answer some specific questions that were answered by the authors in this qualitative analysis.

What were the HCWs included in the supporting staff?

Line 160: [Error! Reference source not found.].????

The answers of the HCWs in the manuscript should be italic to distinguish them from the general text.

Please provide the individual demographic data of each respondent in the form of a table.

It is not clear how the theme 3 statements are related to the blasting myths of vaccination.

The limitation section of the manuscript is not available. There are several limitations in the manuscript i.e. generalizability, bias, etc., Please include them before conclusions

The conclusion section is very simple, this section should be revised according to the major findings and should be connected with the directions for future research and policies.

Author Response

Response to Reviewer 1 Comments

Thank you for the invitation. This is an interesting study. Other quantitative studies could not answer some specific questions that were answered by the authors in this qualitative analysis

Thank you Reviewer 1 for the kind comments. We agree that our qualitative approach complemented the quantitative findings in literature.

What were the HCWs included in the supporting staff?

The supporting staff includes clinic clerks, patient care assistants and phlebotomist as listed on line 150-151.

Line 160: [Error! Reference source not found.].????

This same comment was also noted from Reviewers 2 and 3. There appears to be an erroneous hyperlink in the reviewers’ download. The hyperlink in the original manuscript links correctly to Figure 1. Can the Editor please check?

The answers of the HCWs in the manuscript should be italic to distinguish them from the general text.

The quotes have been edited to italic as suggested.

Please provide the individual demographic data of each respondent in the form of a table.

Taking comments from Reviewer 1 and Reviewer 2 into consideration, Table 2 is replaced with a table that listed individual participant’s demographic data rather than summarised demographic data.

It is not clear how the theme 3 statements are related to the blasting myths of vaccination.

Vaccinations were initially thought by many HCWs to be an imposition, but their actual experience were far more palatable than their assumptions eg. the booking process was simpler and quicker than they imagined. The key myth of vaccination was nonetheless about the side effects which were discussed in subtheme ‘Skepticism towards vaccine’ under theme 1. The quotes in theme 3 showed that side effects experienced by participants were much milder and not as severe as what they were made to believe previously. Clarifications have now been made to section 3.3 to make this clearer.

The limitation section of the manuscript is not available. There are several limitations in the manuscript i.e. generalizability, bias, etc., Please include them before conclusions

The limitation section, which includes generalizability and description of possible bias, is available in the last paragraph in Discussion. We can see how Reviewer 1 could miss it, and so have added a sentence “There are limitations to our study.” at the start of the paragraph to clearly identify this section.

The conclusion section is very simple, this section should be revised according to the major findings and should be connected with the directions for future research and policies.

We have amended the conclusions substantially as suggested. Please let us know whether this section is now adequate.

Reviewer 2 Report

Dear Authors,

your paper entitled “Experience of COVID-19 vaccination among primary healthcare workers in Hong Kong: A qualitative study” is interesting, but I believe that it needs major revisions before it can be considered for possible publication, as for my comments listed here below for the different sections of the manuscript.

Best regards,

the Reviewer.

Abstract

1.       Authors state “Recent studies show coronavirus disease 2019 (COVID-19) vaccine hesitancy”

I don’t know whether this affirmation is correct: e.g. in my country, i.e. Italy, it is not as >95% of HCW are vaccinated against SARS-CoV-2, also because it is required by a specific norm in order to continue their job. My sense is that similar adhesion rates are also reached in other European and North America countries, while I don’t have a full picture of all the world’s countries. Taking anti-flu vaccine in HCW as a reference, in this case the adhesion rates according to published studies were much lower, around 20-30%, with only some peaks of >50 in specific contexts. Of course there is a hesitancy also for anti-Covid-19 vaccine, as there is for anti-flu, both for the general public as well as for HCW. But I believe that the hesitancy is higher for anti-influenza compared to anti-COVID-19, and moreover it is higher in the general public compared to HCW. Please limit the affirmation to the countries/world areas for which you are sure that the affirmation is correct and give proper updated references (not only one, unless a well designed and and with a sufficiently high number of included studies systematic review or an official report of an international institution as WHO) in order to prove your statement in the introduction.

Introduction

2.       Lines 32 and following: you can partially skip this part as I’ve seen many many papers declaring what is SARS-CoV-2 and when it was discovered: we now know all and it is not necessary to explain! Moreover, your statement is a little bit unclear, as the virus was dated back to December 2019, while the first cases in China and an extremely severe epidemic were detected in January, with a spread of the cases during February in many world’s countries. It is true that the pandemic was declared by the WHO on the 11th March 2020, if I correctly remember.

3.       Line 39: a comma missing between “patients” and “particularly”. Moreover, please note that the vaccination is also a fundamental occupational health and safety preventive measure for the protection of the exposed workers, not only of the patients.

4.       You cite here a rapid systematic review dated back to 2021: please consider my comment above made for the abstract. We are in the second part of 2022 now, and I think that many other papers on the vaccines’ hesitancy against COVID-19 came out: please update your personal litarature search and references.

5.         correctdiscovered unfortunately COVID-19 data become old quite fast, and now we have more that 4.7 milion deaths. Please update the data if you can. Moreover, ref. 1 weblink does not work. Perhaps you can consider also WHO: https://covid19.who.int/

6.       Lines 22-24: you refer to extra-UE approvals. Shouldn’t it be appropriate to refer also to AIFA and/or EMA approvals as your study is conducted in Europe?

Methods

7.       Looking at the title of your study and the first descriptions provided in the Methods section it seems that your aim was to describe a situation representative of that of the HCW from Hong-Kong. I don’t know how many HCW there are in honk Kong but I suppose various thousands. Only at the beginning of the results section it is clear that your study is a case analysis of 28 interviews only: please change the title of your article and clearly state the type and design of the study in the first paragraph of the Methods section, which has to be expanded.

8.       The way you have to cite the contribution of KN to the stuidy sounds a little bit not scientific to me in some points, as e.g. when you tell what is KN’s mother tongue. Please refer to similar studies and look how they report their methods.

Results

9.       Table 2: as the number of cases is so low, many rows of the table are not needed: my suggestion is to report standard deviation together with mean age, and to delete age classes. Moreover, for work experience, you can consider ≤10, 11-20 and ≥ 21, instead of <10, 10-19, 20-29 and ≥ 30, saving 1 row with only 1 person represented. Same for the row on “Time between vaccination and launch..:”: the last line “5 or more” can be deleted and included in the previous one to be titled as “4 or more”. The interesting data of the person who waited more than 5 months for the vaccine can be separately reported as text in the results section.

10.   Line 160 of the updated .pdf manuscript I believe there is a problem with the citation of the reference.

11.   Figure 1: according to my previous observation I don’t agree with the title of the figure, as it seesm that your study represents all the HCW of Hong Kong.

Discussion and References

12.   Unfortunately, the discussion and the ref. list, as I highlighted also for the introduction, make me think that you did not check the published scientific literature with a suffient level of attention before writing your study: you cite only 2 works from 2022, while the body of papers published in the latest months on SARS-CoV-2 vaccine hesitancy is very high. Please, this is only an example of what can be found in literature: there are 43 systematic reviews published in 2022 only (other 20 in 2021), while the number of papers published in 2022 on the this topic is of 1,687 to date (more than 3,000 considering 2021). The stream of the PubMed search is: ["COVID-19" AND ("vaccine" OR "Vaccination") AND hesitancy] . I believe that among this vast literature you can find some more similar studies to make comparisons and to be discussed together with your results.

1

Cite Share

Global Predictors of COVID-19 Vaccine Hesitancy: A Systematic Review.

Pires C.

Vaccines (Basel). 2022 Aug 18;10(8):1349. doi: 10.3390/vaccines10081349.

PMID: 36016237 Free PMC article. Review.

BACKGROUND: vaccine hesitancy is defined as a delay in the acceptance or refusal of vaccination, even though immunisation is a determinant in reducing the mortality and morbidity associated with Coronavirus Disease 2019 (COVID-19). AIM: to ident …

2

Cite Share

COVID-19 vaccine hesitancy among pregnant women: a systematic review and meta-analysis.

Bhattacharya O, Siddiquea BN, Shetty A, Afroz A, Billah B.

BMJ Open. 2022 Aug 18;12(8):e061477. doi: 10.1136/bmjopen-2022-061477.

PMID: 35981769 Free PMC article.

Among the participants, only 49% (95% CI 42% to 56%, p<0.001) had COVID-19 vaccine acceptance. High-income countries (47%; 95% CI 38% to 55%, p<0.001), participants with fewer than 12 years of education (38%; 95% CI 19% to 58%, p<0.001) and mu …

3

Cite Share

Prevalence and Determinants of COVID-19 Vaccine Acceptance Among Healthcare Workers: A Systematic Review.

Desye B.

Front Public Health. 2022 Jul 28;10:941206. doi: 10.3389/fpubh.2022.941206. eCollection 2022.

PMID: 35968421 Free PMC article.

Factors such as sex (male), age, profession (medical doctors), and previous influenza vaccination were the main positive predictors for COVID-19 vaccine acceptance among HCWs. ...Therefore, to improve the COVID-19 vaccine acceptanc …

4

Cite Share

Adult vaccination uptake strategies in low- and middle-income countries: A systematic review.

Perroud JM, Soldano S, Avanceña ALV, Wagner A.

Vaccine. 2022 Aug 26;40(36):5313-5321. doi: 10.1016/j.vaccine.2022.07.054. Epub 2022 Aug 8.

PMID: 35953323 Review.

The majority (15/22, 68%) of interventions were multi-component. 82% (18/22) of studies addressed thoughts and feelings, 59% (13/22) addressed social processes, and 73% (16/22) addressed practical issues. Five studies reported primary outcomes of vaccination intent, and th …

5

Cite Share

Prevalence and Determinants of COVID-19 Vaccine Hesitancy Among the Ethiopian Population: A Systematic Review.

Yehualashet DE, Seboka BT, Tesfa GA, Mamo TT, Yawo MN, Hailegebreal S.

Risk Manag Healthc Policy. 2022 Jul 29;15:1433-1445. doi: 10.2147/RMHP.S368057. eCollection 2022.

PMID: 35937966 Free PMC article. Review.

INTRODUCTION: Although vaccination is the most effective way to end the COVID-19 pandemic, there are growing concerns that vaccine hesitancy may undermine its effectiveness. In Ethiopia, vaccine hesitancy forms a major challenge to …

6

Cite Share

Interventions to increase COVID-19 vaccine uptake: a scoping review.

Andreas M, Iannizzi C, Bohndorf E, Monsef I, Piechotta V, Meerpohl JJ, Skoetz N.

Cochrane Database Syst Rev. 2022 Aug 3;8(8):CD015270. doi: 10.1002/14651858.CD015270.

PMID: 35920693 Review.

This scoping review maps interventions aimed at increasing COVID-19 vaccine uptake and decreasing COVID-19 vaccine hesitancy. ...SELECTION CRITERIA: We included studies that assess the impact of interventions implemented to enhance …

7

Cite Share

First COVID-19 Booster Dose in the General Population: A Systematic Review and Meta-Analysis of Willingness and Its Predictors.

Galanis P, Vraka I, Katsiroumpa A, Siskou O, Konstantakopoulou O, Katsoulas T, Mariolis-Sapsakos T, Kaitelidou D.

Vaccines (Basel). 2022 Jul 8;10(7):1097. doi: 10.3390/vaccines10071097.

PMID: 35891260 Free PMC article. Review.

The most important reasons for decline were adverse reactions and discomfort experienced after previous COVID-19 vaccine doses and concerns for serious adverse reactions to COVID-19 booster doses. ...Our findings are innovative and could help po …

8

Cite Share

COVID-19 vaccination hesitancy in pregnant and breastfeeding women and strategies to increase vaccination compliance: a systematic review and meta-analysis.

Bianchi FP, Stefanizzi P, Di Gioia MC, Brescia N, Lattanzio S, Tafuri S.

Expert Rev Vaccines. 2022 Jul 20:1-12. doi: 10.1080/14760584.2022.2100766. Online ahead of print.

PMID: 35818804

INTRODUCTION: Pregnant and breastfeeding women are at an increased risk of severe illness from COVID-19. Despite this, low vaccination coverages are reported in this population sub-group. AREAS COVERED: The purpose of this study is to estimate the proportion …

9

Cite Share

Misinformation About COVID-19 Vaccines on Social Media: Rapid Review.

Skafle I, Nordahl-Hansen A, Quintana DS, Wynn R, Gabarron E.

J Med Internet Res. 2022 Aug 4;24(8):e37367. doi: 10.2196/37367.

PMID: 35816685 Free PMC article. Review.

False claims about adverse vaccine side effects, such as vaccines being the cause of autism, were already considered a threat to global health before the outbreak of COVID-19. ...These studies implied that the misinformation spread on social media had a negat …

10

Cite Share

Safety and Adverse Events Related to COVID-19 mRNA Vaccines; a Systematic Review.

SeyedAlinaghi S, Karimi A, Pashaei Z, Afzalian A, Mirzapour P, Ghorbanzadeh K, Ghasemzadeh A, Dashti M, Nazarian N, Vahedi F, Tantuoyir MM, Shamsabadi A, Dadras O, Mehraeen E.

Arch Acad Emerg Med. 2022 May 22;10(1):e41. doi: 10.22037/aaem.v10i1.1597. eCollection 2022.

PMID: 35765616 Free PMC article.

INTRODUCTION: Knowledge of vaccine-related adverse events is crucial as they are among the most important factors that cause hesitation in receiving vaccines. ...We selected original studies that explored the side effects of mRNA COVID-19 vaccines usin …

11

Cite Share

COVID-19 vaccination hesitancy in Italian healthcare workers: a systematic review and meta-analysis.

Bianchi FP, Stefanizzi P, Brescia N, Lattanzio S, Martinelli A, Tafuri S.

Expert Rev Vaccines. 2022 Jun 30:1-12. doi: 10.1080/14760584.2022.2093723. Online ahead of print.

PMID: 35757890

INTRODUCTION: As for other vaccines, vaccination hesitancy may be a determining factor in the success (or otherwise) of the COVID-19 immunization campaign in healthcare workers (HCWs). AREAS COVERED: To estimate the proportion of HCWs in Italy who expr …

12

Cite Share

The Health Belief Model Applied to COVID-19 Vaccine Hesitancy: A Systematic Review.

Limbu YB, Gautam RK, Pham L.

Vaccines (Basel). 2022 Jun 18;10(6):973. doi: 10.3390/vaccines10060973.

PMID: 35746581 Free PMC article.

Other modifying variables that influenced vaccine hesitancy were knowledge of COVID-19, prior diagnosis of COVID-19, history of flu vaccination, religion, nationality, and political affiliation. The results show that HBM is useful …

13

Cite Share

Hesitancy of COVID-19 vaccines: Rapid systematic review of the measurement, predictors, and preventive strategies.

Anakpo G, Mishi S.

Hum Vaccin Immunother. 2022 Nov 30;18(5):2074716. doi: 10.1080/21645515.2022.2074716. Epub 2022 Jun 17.

PMID: 35714274 Free PMC article.

Vaccine hesitancy is one of the top ten global health threats and the first threat to fighting COVID-19 through vaccination. With the increasing level of COVID-19 vaccine hesitancy amidst the rising level of confirmed …

14

Cite Share

COVID-19 Vaccine Hesitancy in Pakistan: A Mini Review of the Published Discourse.

Khalid S, Usmani BA, Siddiqi S.

Front Public Health. 2022 May 31;10:841842. doi: 10.3389/fpubh.2022.841842. eCollection 2022.

PMID: 35712302 Free PMC article. Review.

This minireview provides a summary of the main findings, features, as well as limitations and gaps in the current epidemiologic research on COVID-19 vaccine hesitancy (VH) in Pakistani population. For this purpose, data on VH studies were extracted fro …

15

Cite Share

COVID-19 Vaccine Acceptance Rate and Its Factors among Healthcare Students: A Systematic Review with Meta-Analysis.

Patwary MM, Bardhan M, Haque MZ, Sultana R, Alam MA, Browning MHEM.

Vaccines (Basel). 2022 May 19;10(5):806. doi: 10.3390/vaccines10050806.

PMID: 35632560 Free PMC article. Review.

(n = 6), China (n = 5), Poland (n = 5), India (n = 2), Italy (n = 2), and Israel (n = 2). The prevalence of the COVID-19 vaccine acceptance rate was 68.8% (95% confidence interval [CI]: 60.8-76.3, I(2) = 100%), and the prevalence of the vaccine hesi …

16

Cite Share

Social media and attitudes towards a COVID-19 vaccination: A systematic review of the literature.

Cascini F, Pantovic A, Al-Ajlouni YA, Failla G, Puleo V, Melnyk A, Lontano A, Ricciardi W.

EClinicalMedicine. 2022 Jun;48:101454. doi: 10.1016/j.eclinm.2022.101454. Epub 2022 May 20.

PMID: 35611343 Free PMC article.

BACKGROUND: Vaccine hesitancy continues to limit global efforts in combatting the COVID-19 pandemic. ...Studies that reported outcomes related to coronavirus disease 2019 (COVID-19) vaccine (attitudes, opinion, etc.) gathered from …

17

Cite Share

COVID-19 vaccine acceptance among health care workers in Africa: A systematic review and meta-analysis.

Ackah M, Ameyaw L, Gazali Salifu M, Afi Asubonteng DP, Osei Yeboah C, Narkotey Annor E, Abena Kwartemaa Ankapong E, Boakye H.

PLoS One. 2022 May 18;17(5):e0268711. doi: 10.1371/journal.pone.0268711. eCollection 2022.

PMID: 35584110 Free PMC article.

Hence, we aimed to review the acceptance rate and possible reasons for COVID-19 vaccine non-acceptance/hesitancy amongst HCWs in Africa. ...The misconceptions and barriers to COVID-19 vaccine acceptance amongst HCWs must be address …

18

Cite Share

Overcoming COVID-19 vaccine hesitancy among ethnic minorities: A systematic review of UK studies.

Hussain B, Latif A, Timmons S, Nkhoma K, Nellums LB.

Vaccine. 2022 May 31;40(25):3413-3432. doi: 10.1016/j.vaccine.2022.04.030. Epub 2022 Apr 28.

PMID: 35534309 Free PMC article. Review.

Ethnic minority communities in the UK have been disproportionately affected by the pandemic, with increased risks of infection, severe disease, and death. Hesitancy around the COVID-19 vaccine may be contributing to disparities in vaccine delive …

19

Cite Share

Pfizer-BioNTech COVID-19 Vaccine (BNT162b2) Side Effects: A Systematic Review.

Dighriri IM, Alhusayni KM, Mobarki AY, Aljerary IS, Alqurashi KA, Aljuaid FA, Alamri KA, Mutwalli AA, Maashi NA, Aljohani AM, Alqarni AM, Alfaqih AE, Moazam SM, Almutairi MN, Almutairi AN.

Cureus. 2022 Mar 26;14(3):e23526. doi: 10.7759/cureus.23526. eCollection 2022 Mar.

PMID: 35494952 Free PMC article. Review.

Vaccinations prevented severe clinical complications of COVID-19. It was considered a vital component of living endemically with COVID-19. ...A total of 107 PubMed and Google Scholar publications were screened for Pfizer-BioNTech COVID-19 …

20

Cite Share

A meta-analysis of COVID-19 vaccine attitudes and demographic characteristics in the United States.

Dhanani LY, Franz B.

Public Health. 2022 Jun;207:31-38. doi: 10.1016/j.puhe.2022.03.012. Epub 2022 Mar 28.

PMID: 35486981 Free PMC article. Review.

OBJECTIVES: Despite the potential for COVID-19 vaccination to prevent severe disease and death, vaccine hesitancy is common in the United States, with more than a quarter of eligible Americans yet to receive the first dose. We draw on existing p …

21

Cite Share

Systematic review of health and disease in Ukrainian children highlights poor child health and challenges for those treating refugees.

Ludvigsson JF, Loboda A.

Acta Paediatr. 2022 Jul;111(7):1341-1353. doi: 10.1111/apa.16370. Epub 2022 Apr 27.

PMID: 35466444 Free PMC article.

Alcohol consumption was common in women of reproductive age, including during pregnancy, risking foetal alcohol syndrome. Neonatal screening programmes provided low coverage. Vaccine hesitancy was common and vaccination rates were low. Other concerns were mea …

22

Cite Share

Effect of COVID-19 Pandemic on Influenza Vaccination Intention: A Meta-Analysis and Systematic Review.

Kong G, Lim NA, Chin YH, Ng YPM, Amin Z.

Vaccines (Basel). 2022 Apr 13;10(4):606. doi: 10.3390/vaccines10040606.

PMID: 35455354 Free PMC article. Review.

Poorer outcomes have been reported with COVID-19 and influenza coinfections. As the COVID-19 pandemic rages on, protection against influenza by vaccination is becoming increasingly important. ...A literature search was conducted on Embase, PubMe …

23

Cite Share

COVID-19 vaccination acceptance among dental students and dental practitioners: A systematic review and meta-analysis.

Lin GSS, Lee HY, Leong JZ, Sulaiman MM, Loo WF, Tan WW.

PLoS One. 2022 Apr 19;17(4):e0267354. doi: 10.1371/journal.pone.0267354. eCollection 2022.

PMID: 35439274 Free PMC article.

BACKGROUND: Dental practitioners and dental students are classified as high-risk exposure to COVID-19 due to the nature of dental treatments, but evidence of their acceptance towards COVID-19 vaccination is still scarce. Hence, this systemic rev …

24

Cite Share

Attitudes of COVID-19 vaccination among college students: A systematic review and meta-analysis of willingness, associated determinants, and reasons for hesitancy.

Geng H, Cao K, Zhang J, Wu K, Wang G, Liu C.

Hum Vaccin Immunother. 2022 Nov 30;18(5):2054260. doi: 10.1080/21645515.2022.2054260. Epub 2022 Apr 19.

PMID: 35438612 Free PMC article.

The significance of COVID-19 vaccine has been declared and this study synthesizes the attitudes and determinants in vaccination hesitancy of college students. ...Proportion and OR with 95% CI were pooled to estimate the acceptance rates and dete …

25

Cite Share

Safety of SARS-CoV-2 vaccination in patients with inflammatory bowel disease: A systematic review and meta-analysis.

James D, Jena A, Bharath PN, Choudhury A, Singh AK, Sebastian S, Sharma V.

Dig Liver Dis. 2022 Jun;54(6):713-721. doi: 10.1016/j.dld.2022.03.005. Epub 2022 Mar 22.

PMID: 35382972 Free PMC article. Review.

INTRODUCTION: Risk of adverse effects and flare of inflammatory bowel disease (IBD) are frequently cited reasons for COVID-19 vaccine hesitancy. METHODS: Electronic databases were searched to identify studies reporting the use of COVID-19 …

26

Cite Share

COVID-19 vaccines and patients with multiple sclerosis: willingness, unwillingness and hesitancy: a systematic review and meta-analysis.

Yazdani A, Mirmosayyeb O, Ghaffary EM, Hashemi MS, Ghajarzadeh M.

Neurol Sci. 2022 Jul;43(7):4085-4094. doi: 10.1007/s10072-022-06051-6. Epub 2022 Apr 5.

PMID: 35381877 Free PMC article. Review.

OBJECTIVE: The purpose of this study was to determine the pooled prevalence of vaccination willingness, unwillingness, and hesitancy among patients with multiple sclerosis. ...Hesitancy pooled prevalence to vaccination among patients with MS was 0% (I( …

27

Cite Share

Antecedents and consequences of COVID-19 conspiracy beliefs: A systematic review.

van Mulukom V, Pummerer LJ, Alper S, Bai H, Čavojová V, Farias J, Kay CS, Lazarevic LB, Lobato EJC, Marinthe G, Pavela Banai I, Šrol J, Žeželj I.

Soc Sci Med. 2022 May;301:114912. doi: 10.1016/j.socscimed.2022.114912. Epub 2022 Mar 14.

PMID: 35354105 Free PMC article. Review.

RATIONALE: Belief in COVID-19 conspiracy theories can have severe consequences; it is therefore crucial to understand this phenomenon, in its similarities with general conspiracy belief, but also in how it is context-dependent. OBJECTIVE: The aim of this systematic …

28

Cite Share

COVID-19 Vaccine Acceptance among Low- and Lower-Middle-Income Countries: A Rapid Systematic Review and Meta-Analysis.

Patwary MM, Alam MA, Bardhan M, Disha AS, Haque MZ, Billah SM, Kabir MP, Browning MHEM, Rahman MM, Parsa AD, Kabir R.

Vaccines (Basel). 2022 Mar 11;10(3):427. doi: 10.3390/vaccines10030427.

PMID: 35335059 Free PMC article. Review.

Widespread vaccination against COVID-19 is critical for controlling the pandemic. ...Being male and perceiving risk of COVID-19 infection were predictors for willingness to accept the vaccine. ...

29

Cite Share

The Effectiveness of Interventions for Increasing COVID-19 Vaccine Uptake: A Systematic Review.

Batteux E, Mills F, Jones LF, Symons C, Weston D.

Vaccines (Basel). 2022 Mar 3;10(3):386. doi: 10.3390/vaccines10030386.

PMID: 35335020 Free PMC article. Review.

Vaccination is vital to protect the public against COVID-19. The aim of this systematic review is to identify and evaluate the effectiveness of interventions to increase COVID-19 vaccine uptake. We searched a range of databases (Embase, M …

30

Cite Share

Potential factors influencing COVID-19 vaccine acceptance and hesitancy: A systematic review.

Roy DN, Biswas M, Islam E, Azam MS.

PLoS One. 2022 Mar 23;17(3):e0265496. doi: 10.1371/journal.pone.0265496. eCollection 2022.

PMID: 35320309 Free PMC article.

BACKGROUND AND AIMS: Although vaccines are considered the most effective and fundamental therapeutic tools for consistently preventing the COVID-19 disease, worldwide vaccine hesitancy has become a widespread public health issue for successful immuniza …

31

Cite Share

Public acceptability of COVID-19 vaccines and its predictors in Middle ‎Eastern/North African (MENA) countries: a systematic review‎.

Dadras O, SeyedAlinaghi S, Karimi A, Shamsabadi A, Mahdiabadi S, Mohammadi P, Amiri A, Shojaei A, Pashaei Z, Mirzapour P, Qaderi K, MohsseniPour M, Alilou S, Mehraeen E, Jahanfar S.

Hum Vaccin Immunother. 2022 Nov 30;18(5):2043719. doi: 10.1080/21645515.2022.2043719. Epub 2022 Mar 23.

PMID: 35318872 Free PMC article.

INTRODUCTION: COVID-19 vaccines emerged as a worldwide hope to contain the pandemic. However, many people are still hesitant to receive these vaccines. We aimed to systematically review the public knowledge, perception, and acceptability of COVID-19 …

32

Cite Share

COVID-19 vaccine acceptance and its associated factors in Ethiopia: A systematic review and meta-analysis.

Mekonnen BD, Mengistu BA.

Clin Epidemiol Glob Health. 2022 Mar-Apr;14:101001. doi: 10.1016/j.cegh.2022.101001. Epub 2022 Mar 7.

PMID: 35284688 Free PMC article. Review.

However, vaccine hesitation is increasing and hindering efforts targeting to reduce the burden of the COVID-19 disease. Hence, determining COVID-19 vaccine acceptance and identifying determinants that would hinder people to vaccina …

33

Cite Share

COVID-19 vaccination among pregnant people in the United States: a systematic review.

Rawal S, Tackett RL, Stone RH, Young HN.

Am J Obstet Gynecol MFM. 2022 Jul;4(4):100616. doi: 10.1016/j.ajogmf.2022.100616. Epub 2022 Mar 10.

PMID: 35283351 Free PMC article. Review.

However, pregnant people were not included in vaccine trials, and there are limited data on COVID-19 vaccines during pregnancy. ...COVID-19 vaccine acceptance was low among pregnant people in the United States. ...

34

Cite Share

Parents' and Guardians' Willingness to Vaccinate Their Children against COVID-19: A Systematic Review and Meta-Analysis.

Chen F, He Y, Shi Y.

Vaccines (Basel). 2022 Jan 24;10(2):179. doi: 10.3390/vaccines10020179.

PMID: 35214638 Free PMC article. Review.

COVID-19 vaccination for children is crucial to achieve herd immunity. This is the first systematic review and meta-analysis to estimate parents' and guardians' willingness to vaccinate their children against COVID-19 and identify the determinan …

35

Cite Share

Ethnic/racial minorities' and migrants' access to COVID-19 vaccines: A systematic review of barriers and facilitators.

Abba-Aji M, Stuckler D, Galea S, McKee M.

J Migr Health. 2022;5:100086. doi: 10.1016/j.jmh.2022.100086. Epub 2022 Feb 18.

PMID: 35194589 Free PMC article. Review.

BACKGROUND: There are widespread concerns that ethnic minorities and migrants may have inadequate access to COVID-19 vaccines. . Improving vaccine uptake among these vulnerable groups is important towards controlling the spread of COVID-19 and r …

36

Cite Share

Willingness, refusal and influential factors of parents to vaccinate their children against the COVID-19: A systematic review and meta-analysis.

Galanis P, Vraka I, Siskou O, Konstantakopoulou O, Katsiroumpa A, Kaitelidou D.

Prev Med. 2022 Apr;157:106994. doi: 10.1016/j.ypmed.2022.106994. Epub 2022 Feb 18.

PMID: 35183597 Free PMC article. Review.

The main predictors of parents' intention to vaccinate their children were fathers, older age of parents, higher income, higher levels of perceived threat from the COVID-19, and positive attitudes towards vaccination (e.g. children's complete vaccination …

37

Cite Share

Determination of Novel Coronavirus Disease (COVID-19) Vaccine Hesitancy Using a Systematic Review Approach Based on the Scientific Articles in PubMed Database.

Ergün A, Bekar A, Aras B, Dere C, Tekneci D, Sarıçiçek G, Akdere SN, Telli S, Pehlivanlı ŞB, Özyurek Ucael D, Özden ME, Altıntaş E, Aslan D.

Turk Thorac J. 2022 Jan;23(1):70-84. doi: 10.5152/TurkThoracJ.2022.21082.

PMID: 35110204

The keyword "COVID-19" is used in 61 articles (79.2%). The second most frequently used keyword is "vaccine hesitancy" (n = 37, 48.1%), followed by "vaccine" (n = 25, 32.5%). ...CONCLUSION: This study illustrates the recent situation for the coro …

38

Cite Share

Global COVID-19 Vaccine Acceptance: A Systematic Review of Associated Social and Behavioral Factors.

Shakeel CS, Mujeeb AA, Mirza MS, Chaudhry B, Khan SJ.

Vaccines (Basel). 2022 Jan 12;10(1):110. doi: 10.3390/vaccines10010110.

PMID: 35062771 Free PMC article. Review.

COVID-19 vaccines have met varying levels of acceptance and hesitancy in different parts of the world, which has implications for eliminating the COVID-19 pandemic. ...Furthermore, poor influenza-vaccination history, as well as conspiracy …

39

Cite Share

Frequency of Adverse Events in the Placebo Arms of COVID-19 Vaccine Trials: A Systematic Review and Meta-analysis.

Haas JW, Bender FL, Ballou S, Kelley JM, Wilhelm M, Miller FG, Rief W, Kaptchuk TJ.

JAMA Netw Open. 2022 Jan 4;5(1):e2143955. doi: 10.1001/jamanetworkopen.2021.43955.

PMID: 35040967 Free PMC article.

IMPORTANCE: Adverse events (AEs) after placebo treatment are common in randomized clinical drug trials. Systematic evidence regarding these nocebo responses in vaccine trials is important for COVID-19 vaccination worldwide especially because concern ab …

40

Cite Share

Cardiovascular and haematological events post COVID-19 vaccination: A systematic review.

Al-Ali D, Elshafeey A, Mushannen M, Kawas H, Shafiq A, Mhaimeed N, Mhaimeed O, Mhaimeed N, Zeghlache R, Salameh M, Paul P, Homssi M, Mohammed I, Narangoli A, Yagan L, Khanjar B, Laws S, Elshazly MB, Zakaria D.

J Cell Mol Med. 2022 Feb;26(3):636-653. doi: 10.1111/jcmm.17137. Epub 2021 Dec 29.

PMID: 34967105 Free PMC article.

Since COVID-19 took a strong hold around the globe causing considerable morbidity and mortality, a lot of effort was dedicated to manufacturing effective vaccines against SARS-CoV-2. ...This review was undertaken with the aim of putting together all the reported car …

41

Cite Share

COVID-19 Disparities and Vaccine Hesitancy in Black Americans: What Ethical Lessons Can Be Learned?

Restrepo N, Krouse HJ.

Otolaryngol Head Neck Surg. 2022 Jun;166(6):1147-1160. doi: 10.1177/01945998211065410. Epub 2021 Dec 14.

PMID: 34905417

DATA SOURCES: An internet search through PubMed, CINAHL, and socINDEX was conducted to identify articles on COVID-19 vaccine hesitation among the Black population between 2020 and 2021. REVIEW METHODS: A systematic review approach was taken to search a …

42

Cite Share

What factors promote vaccine hesitancy or acceptance during pandemics? A systematic review and thematic analysis.

Truong J, Bakshi S, Wasim A, Ahmad M, Majid U.

Health Promot Int. 2022 Feb 17;37(1):daab105. doi: 10.1093/heapro/daab105.

PMID: 34244738

Examine the factors that promote vaccine hesitancy or acceptance during pandemics, major epidemics and global outbreaks. ...An understanding of participant experiences and perspectives toward vaccines from previous pandemics will greatly inform the development of st …

43

Cite Share

COVID-19 vaccination intention in the first year of the pandemic: A systematic review.

Al-Amer R, Maneze D, Everett B, Montayre J, Villarosa AR, Dwekat E, Salamonson Y.

J Clin Nurs. 2022 Jan;31(1-2):62-86. doi: 10.1111/jocn.15951. Epub 2021 Jul 6.

PMID: 34227179 Free PMC article. Review.

BACKGROUND: As COVID-19 vaccine becomes available worldwide, attention is being directed to community vaccine uptake, to achieve population-wide immunity. ...Findings highlighted that socio-demographic differences, perceptions of risk and susceptibilit …

13.   Please expand the limitation section of your discussion, considering the intrinsic limitations of your study design.

Conclusions

14.   Please, consider that your conclusions have to be a direct consequence of your results, and that the data you are describing is not representative of the Hong Kong situation.

Author Response

Response to Reviewer 2 Comments

Your paper entitled “Experience of COVID-19 vaccination among primary healthcare workers in Hong Kong: A qualitative study” is interesting, but I believe that it needs major revisions before it can be considered for possible publication, as for my comments listed here below for the different sections of the manuscript.

Thank you Reviewer 2 for the forthrightness. We hope our responses to Reviewer 2’s comments are satisfactory.

Abstract

  1. Authors state “Recent studies show coronavirus disease 2019 (COVID-19) vaccine hesitancy”

I don’t know whether this affirmation is correct: e.g. in my country, i.e. Italy, it is not as >95% of HCW are vaccinated against SARS-CoV-2, also because it is required by a specific norm in order to continue their job. My sense is that similar adhesion rates are also reached in other European and North America countries, while I don’t have a full picture of all the world’s countries. Taking anti-flu vaccine in HCW as a reference, in this case the adhesion rates according to published studies were much lower, around 20-30%, with only some peaks of >50 in specific contexts. Of course there is a hesitancy also for anti-Covid-19 vaccine, as there is for anti-flu, both for the general public as well as for HCW. But I believe that the hesitancy is higher for anti-influenza compared to anti-COVID-19, and moreover it is higher in the general public compared to HCW. Please limit the affirmation to the countries/world areas for which you are sure that the affirmation is correct and give proper updated references (not only one, unless a well designed and and with a sufficiently high number of included studies systematic review or an official report of an international institution as WHO) in order to prove your statement in the introduction.

We appreciate Reviewer 2’s comment, and we have dived a little bit more into the literature. A recently published (Aug 2022) scoping review of 26 papers (22 cross-sectional surveys, 2 cross-sectional qualitative reports, a commentary and a systematic review), that looked into the intentions and attitudes on COVID-19 vaccination of 43,199 HCWs in 16 different countries in Africa, Asia, Europe (including Italy) and North America, found that vaccine hesitancy does exist in HCWs albeit at varying degrees in different populations. Of the papers that compared HCWs with the general population, 2 papers mentioned significantly higher vaccination willingness for HCWs than the general population while 2 found no statistically significant difference, and one found that doctors and the general population were more willing to get vaccinated than nurses. [Reference: Willems LD, Dyzel V, Sterkenburg PS. COVID-19 Vaccination Intentions amongst Healthcare Workers: A Scoping Review. Int J Environ Res Public Health [Internet]. 2022 Aug 17 [cited 2022 Sep 2];19(16):10192. Available from https://www.ncbi.nlm.nih.gov/pmc/articles/PMC9407815/]

Hence, we have now amended the first sentence in the abstract to “Studies show coronavirus disease 2019 (COVID-19) vaccine hesitancy exists among healthcare workers (HCWs).” ie. deleting the mention of the general population.

We have also added the findings from Willems et al’s scoping review to the introduction as suggested by the reviewer in comment no. 4.

Introduction

  1. Lines 32 and following: you can partially skip this part as I’ve seen many many papers declaring what is SARS-CoV-2 and when it was discovered: we now know all and it is not necessary to explain! Moreover, your statement is a little bit unclear, as the virus was dated back to December 2019, while the first cases in China and an extremely severe epidemic were detected in January, with a spread of the cases during February in many world’s countries. It is true that the pandemic was declared by the WHO on the 11thMarch 2020, if I correctly remember.

We have deleted the first sentence on line 32 as suggested and adjusted the sentence to assist with flow.

  1. Line 39: a comma missing between “patients” and “particularly”. Moreover, please note that the vaccination is also a fundamental occupational health and safety preventive measure for the protection of the exposed workers, not only of the patients.

Comma added. And we agree that vaccination is for the protection of HCWs as well as patients. We have added the mention of the HCWs in the sentence on line 41.

  1. You cite here a rapid systematic review dated back to 2021: please consider my comment above made for the abstract. We are in the second part of 2022 now, and I think that many other papers on the vaccines’ hesitancy against COVID-19 came out: please update your personal litarature search and references.

Please see our response to comment no. 1. Reference 4 is now replaced with another review published in Aug 2022, and the text has been updated accordingly.

  1. correctdiscovered unfortunately COVID-19 data become old quite fast, and now we have more that 4.7 milion deaths. Please update the data if you can. Moreover, ref. 1 weblink does not work. Perhaps you can consider also WHO: https://covid19.who.int/

We believe this is a ‘copy and paste’ error by Reviewer 2. We did not reference any death statistics in the main text and reference 1 does not contain a weblink. We will be happy to respond accordingly if Reviewer 2 could clarify.

  1. Lines 22-24: you refer to extra-UE approvals. Shouldn’t it be appropriate to refer also to AIFA and/or EMA approvals as your study is conducted in Europe?

Again, we believe this is a ‘copy and paste’ error. Lines 22-24 in the Abstract do not refer to any approval. And again, we will be happy to respond accordingly if Reviewer 2 could clarify.

Methods

  1. Looking at the title of your study and the first descriptions provided in the Methods section it seems that your aim was to describe a situation representative of that of the HCW from Hong-Kong. I don’t know how many HCW there are in honk Kong but I suppose various thousands. Only at the beginning of the results section it is clear that your study is a case analysis of 28 interviews only: please change the title of your article and clearly state the type and design of the study in the first paragraph of the Methods section, which has to be expanded.

We stated very clearly in our title that this is a qualitative study. And we stated very clearly in the abstract that 28 HCWs were interviewed. Our phenomenological study approach and interview design, not case analysis, are also stated very clearly in Methods (section 2.1). Qualitative research is not dependent on sample size. Our sample of 28 participants have contributed data that reached data saturation which is the key end point in qualitative research. We feel that our title and the first paragraph of the Methods section are adequate.

  1. The way you have to cite the contribution of KN to the study sounds a little bit not scientific to me in some points, as e.g. when you tell what is KN’s mother tongue. Please refer to similar studies and look how they report their methods.

We disagree with the reviewer. We used the consolidated criteria for reporting qualitative studies (COREQ) checklist to guide the writing of this manuscript. A key component of qualitative research reporting is to describe the researchers, their roles and their relationship with their participants. Not only is this scientific, it is a demonstration of rigour and transparency of the processes in qualitative research. Stating that KN used his mother tongue to conduct the interviews with participants with the same mother tongue gives readers a sense of the rapport between interviewer and interviewees. Author PL has written a similar report stating the use of the interviewer’s native language with their focus group participants. [Saberi S, Wachtler C, Lau P. Are we on the same page? Mental health literacy and access to care: a qualitative study in young Hazara refugees in Melbourne. Aust J Prim Health. 2021 Dec;27(6):450-455. doi: 10.1071/PY21017. PMID: 34802509.]

Reference: Allison Tong, Peter Sainsbury, Jonathan Craig, Consolidated criteria for reporting qualitative research (COREQ): a 32-item checklist for interviews and focus groups, International Journal for Quality in Health Care, Volume 19, Issue 6, December 2007, Pages 349–357, https://doi.org/10.1093/intqhc/mzm042

Results

  1. Table 2: as the number of cases is so low, many rows of the table are not needed: my suggestion is to report standard deviation together with mean age, and to delete age classes. Moreover, for work experience, you can consider ≤10, 11-20 and ≥ 21, instead of <10, 10-19, 20-29 and ≥ 30, saving 1 row with only 1 person represented. Same for the row on “Time between vaccination and launch..:”: the last line “5 or more” can be deleted and included in the previous one to be titled as “4 or more”. The interesting data of the person who waited more than 5 months for the vaccine can be separately reported as text in the results section.

As explained in our response to comment no. 7, our sample size is not low but is adequate for qualitative research. We have added another table to describe the demographics of individual participants as per the request of Reviewer 1. We feel that the added table is clearer and more adequately provide context, and so have now replaced the original Table 2 with the new Table.

  1. Line 160 of the updated .pdf manuscript I believe there is a problem with the citation of the reference.

This same comment was also from Reviewers 1 and 3. There appears to be an erroneous hyperlink in the reviewers’ download. The hyperlink in the original manuscript links correctly to Figure 1. Can the Editor please check?

  1. Figure 1: according to my previous observation I don’t agree with the title of the figure, as it seesm that your study represents all the HCW of Hong Kong.

        We disagree with the reviewer. Figure 1 is the diagrammatical representation of the “themes and subthemes of the study” which is the title. It does not say that the findings represent all HCWs in Hong Kong.

Discussion and References

  1. Unfortunately, the discussion and the ref. list, as I highlighted also for the introduction, make me think that you did not check the published scientific literature with a suffient level of attention before writing your study: you cite only 2 works from 2022, while the body of papers published in the latest months on SARS-CoV-2 vaccine hesitancy is very high. Please, this is only an example of what can be found in literature: there are 43 systematic reviews published in 2022 only (other 20 in 2021), while the number of papers published in 2022 on the this topic is of 1,687 to date (more than 3,000 considering 2021). The stream of the PubMed search is: ["COVID-19" AND ("vaccine" OR "Vaccination") AND hesitancy] . I believe that among this vast literature you can find some more similar studies to make comparisons and to be discussed together with your results.

        Thank you Reviewer 2 for the suggestion. We performed an updated literature search as suggested and updated the references in our Discussion. A new study added as reference 17 which was published in Jan 2022 to support action cues motivating HCWs for vaccination from line 389 to 392. The finding of HCWs fearing of vaccine side effects is now replaced with another paper published in Apr 2022 from line 397 to 399.

  1. Please expand the limitation section of your discussion, considering the intrinsic limitations of your study design.

Recall bias is now mentioned in the limitation section (last paragraph) of Discussion.

Conclusions

  1. Please, consider that your conclusions have to be a direct consequence of your results, and that the data you are describing is not representative of the Hong Kong situation.

We have revised the conclusions substantially in response to Reviewer 1’s suggestion to connect major findings “with the directions for future research and policies”. We reiterate that the aim of qualitative research is not to seek ‘representativeness’ of the study population. Our study approach was to explore and gain an understanding of the ‘phenomenon’ of vaccine hesitation of HCWs in Hong Kong, and our participants were selected to achieve maximum variation to ensure that we have broad representation of views (see Section 2.2). Therefore, within the limitations we have already described in the last paragraph in Discussion, we believe our methods and processes were rigorous and valid to justify our conclusions.

Reviewer 3 Report

This is a qualitative study to evaluate the perception of healthcare workers in Hing Kong on the SARS-CoV-2 vaccination. The article provides us with relevant information when we understand healthcare workers’ thoughts on the SARS-CoV-2 vaccination from a global point of view.

Major)

1) The authors mentioned that a number of participants’ decisions to vaccinate were heavily influenced by government policies or organizational regulatory measures (Line 403). On the other hand, the study identified ‘autonomy’ as a subtheme (Line 205). It would be appreciated if the authors added participant comments on ensuring self-decision making.

2) There might be a limitation on recall bias, which should be briefly mentioned in the discussion.

3) There would be differences in the perception, which was classified in theme ‘1’, between junior and senior healthcare workers.

Minor) reference error in line 160

Overall, the article is well described; however, minor modification is necessary, as mentioned above.

Author Response

Response to Reviewer 3 Comments

This is a qualitative study to evaluate the perception of healthcare workers in Hing Kong on the SARS-CoV-2 vaccination. The article provides us with relevant information when we understand healthcare workers’ thoughts on the SARS-CoV-2 vaccination from a global point of view.

Thank you Reviewer 3 for the kind comment.

Major)

1) The authors mentioned that a number of participants’ decisions to vaccinate were heavily influenced by government policies or organizational regulatory measures (Line 403). On the other hand, the study identified ‘autonomy’ as a subtheme (Line 205). It would be appreciated if the authors added participant comments on ensuring self-decision making.

Whilst the Hong Kong government actively promotes and encourages vaccination, there are in fact no mandatory measures enforced in Hong Kong, and HCPs and the general public have the rights to make their own decision on vaccination. In the current quote under the subtheme ‘autonomy’, the participant was in fact emphasizing the autonomy they have and describing their self-decision making. We have amended the sentence preceding the quote to make this clearer – “Participants valued the autonomy that they currently have and believed their decisions about whether to vaccinate should be respected.”

To further clarify, we have added “There are currently no mandatory measures enforced in Hong Kong, and HCWs and the general public have their rights to decide whether to vaccinate.” on line 421 under Discussion.

2) There might be a limitation on recall bias, which should be briefly mentioned in the discussion.

Thank you Reviewer 3 for the suggestion. Recall bias is now added to the last paragraph of Discussion.

3) There would be differences in the perception, which was classified in theme ‘1’, between junior and senior healthcare workers.

We agree with Reviewer 3 and this was in fact what we found. We have now added a sentence “In particular, junior HCWs such as supporting staff and some nurses seemed more likely to have such skepticisms.” under subtheme ‘Skepticism towards vaccine’.

Minor) reference error in line 160

This same comment was also from Reviewers 1 and 2. There appears to be an erroneous hyperlink in the reviewers’ download. The hyperlink in the original manuscript links correctly to Figure 1. Can the Editor please check?

Round 2

Reviewer 2 Report

Dear Authors,

your article has been improved after the peer review process and I believe that it is now ready for publication in Vaccines journal.

Best regards,

the Reviewer